# Identification and Regulation of Hypoxia-Tolerant and Germination-Related Genes in Rice

**DOI:** 10.3390/ijms25042177

**Published:** 2024-02-11

**Authors:** Hongyan Yuan, Zhenzhen Zheng, Yaling Bao, Xueyu Zhao, Jiaqi Lv, Chenghang Tang, Nansheng Wang, Zhaojie Liang, Hua Li, Jun Xiang, Yingzhi Qian, Yingyao Shi

**Affiliations:** 1College of Agronomy, Anhui Agricultural University, Hefei 230036, China; 20721063@stu.ahau.edu.cn (H.Y.); zzz4560902@163.com (Z.Z.); valleybao2019@126.com (Y.B.); 19990114@stu.ahau.edu.cn (X.Z.); 23721081@stu.ahau.edu.cn (J.L.); chsoup@stu.ahau.edu.cn (C.T.); 1341489953@stu.ahau.edu.cn (N.W.); 21721653@stu.ahau.edu.cn (Z.L.); lh2550361252@163.com (H.L.); xj1259006818@stu.ahau.edu.cn (J.X.); 21720172@stu.ahau.edu.cn (Y.Q.); 2Institute of Crop Sciences, Chinese Academy of Agricultural Sciences, Beijing 100081, China

**Keywords:** rice, direct seeding, hypoxia-tolerant germination, genes, regulation

## Abstract

In direct seeding, hypoxia is a major stress faced by rice plants. Therefore, dissecting the response mechanism of rice to hypoxia stress and the molecular regulatory network is critical to the development of hypoxia-tolerant rice varieties and direct seeding of rice. This review summarizes the morphological, physiological, and ecological changes in rice under hypoxia stress, the discovery of hypoxia-tolerant and germination-related genes/QTLs, and the latest research on candidate genes, and explores the linkage of hypoxia tolerance genes and their distribution in indica and japonica rice through population variance analysis and haplotype network analysis. Among the candidate genes, *OsMAP1* is a typical gene located on the MAPK cascade reaction for indica–japonica divergence; MHZ6 is involved in both the MAPK signaling and phytohormone transduction pathway. *MHZ6* has three major haplotypes and one rare haplotype, with Hap3 being dominated by indica rice varieties, and promotes internode elongation in deep-water rice by activating the *SD1* gene. *OsAmy3D* and Adh1 have similar indica–japonica varietal differentiation, and are mainly present in indica varieties. There are three high-frequency haplotypes of *OsTPP7*, namely Hap1 (n = 1109), Hap2 (n = 1349), and Hap3 (n = 217); Hap2 is more frequent in japonica, and the genetic background of *OsTPP7* was derived from the japonica rice subpopulation. Further artificial selection, natural domestication, and other means to identify more resistance mechanisms of this gene may facilitate future research to breed superior rice cultivars. Finally, this study discusses the application of rice hypoxia-tolerant germplasm in future breeding research.

## 1. Introduction

Rice (*Oryza sativa* L.) is one of the most important food crops in the world. Currently, direct seeding of rice has become a popular agricultural practice for its labor-saving, clean, and efficient characteristics [1]. Moreover, it shows obvious yield advantages over transplanting rice for its significantly higher number of grains per spike, number of spikes per unit area, and thousand-grain weight (TGW) [2]. However, direct seeding of rice is also more easily affected by adverse environmental factors and weather in the field than traditional cultivation patterns, particularly hypoxia stress. Hypoxic germination involves a series of morphological, physiological, biochemical, and metabolic changes [3]. Direct-seeded rice with a lower capability of hypoxic germination tends to exhibit low seedling completeness and robustness, which ultimately affects the yield [4]. Previous studies have demonstrated that under hypoxic stress, hypoxia-tolerant varieties can rapidly elongate their coleoptiles to protrude from the water surface so as to obtain sufficient oxygen to ensure normal life activities. Yu Yang et al. [5] utilized 1126 genotypes materials to observe hypoxia traits at a water depth of 5 cm. The results revealed that hypoxia-tolerant materials showed rapid coleoptile elongation on day 6 and broke through the water surface to obtain sufficient oxygen on day 8–9, whereas the non-tolerant materials showed stagnation after rapid growth, and ultimately died due to the lack of oxygen.

Under continuous domestication, rice has developed two strategies to cope with hypoxic stress: the low-oxygen escape strategy (LOES) and the low-oxygen quiescent strategy (LOQS) [6]. In addition to coleoptiles, the growth and development of roots are also affected by the availability of oxygen. When the coleoptile penetrates the water surface and comes into contact with the air, the root will begin to grow and develop rapidly due to the availability of sufficient oxygen. Kawai et al. [7] found that root development was inhibited when rice seedlings were continuously immersed in water for 7 days, but would be restored once the seedlings were transferred to normal conditions. Under hypoxic stress, the coleoptile will undergo rapid elongation and root development will be delayed, indicating that the nutrients and energy are redistributed and preferentially supplied to the coleoptile to promote its growth.

Phytohormones play very important roles in plant growth and development and responses to adverse stress, among which auxin and ethylene are important hormones affecting hypoxic germination [8,9,10]. miRNAs play an array of important roles in plant growth, development, and metabolism, along with involvement in abiotic stress and pathogen responses [11]. miRNAs have been reported as key regulators of plant root development architecture via targeting AUXIN RESPONSE TRANSCRIPTION FACTOR (ARFs) [12]. MicroRNAs play important roles in stressful environments in *Arabidopsis thaliana*, maize, rice, and other crops and help to improve crop stress tolerance, and the expression levels of miR156g, miR157d, miR158a, miR159a, miR172a,b, miR391, and miR775 were enhanced in *Arabidopsis* roots under low-oxygen stress [13]. Taking advantage of high-throughput small RNA sequencing, a group of microRNAs were identified in maize and teosinte under two crucial environmental stresses, submergence and drought, and alternated stresses [14]. In rice, miR393 can degrade miRNAs encoding the auxin receptor, which in turn reduces the expression of auxin response factor (ARF) [15]. Under hypoxic stress, miR393 expression is inhibited, which will enhance the expression of ARF, activate the auxin signaling pathway, and ultimately promote the elongation and growth of coleoptiles [9]. miR393 is encoded by two loci, OsMIR393a and OsMIR393b [16,17], and OsMIR393b is expressed in shoot meristematic tissues, coleoptiles, and stomatal cells [9]. In addition, miR393 plays important roles in regulating responses to drought stress, low-temperature stress, metal stress, and pest and disease attack in rice [18,19,20,21]. Sasidharan et al. [22] found that the endogenous ethylene level increased rapidly under flooded conditions, which would inhibit root growth. When the coleoptile grows rapidly to penetrate the water surface to obtain sufficient oxygen, the ethylene level will decrease, which further promotes root growth and development. Ethylene-responsive transcription factors (ERFs) control genes encoding anaerobic metabolism, as well as up-regulate the gibberellin (GA) biosynthesis gene *GA30x*.

Recently, it was found that ethylene plays a crucial role in flooding response and metabolic acclimatization of rice, and ethylene improves rice survival at the seedling stage by enhancing specific transcription factors (VII ethylene response factors), probably by inducing autophagy and reactive oxygen species (ROS) production [23]. In recent years, more and more QTLs and candidate genes for hypoxic germination tolerance have been discovered and cloned, such as *OsTPP7*, *OsHIGD2*, and *Sub1* [24,25,26,27,28,29,30,31,32,33,34]. Therefore, understanding the genes/QTL affecting rice seed germination and emergence under hypoxia stress and their roles in the mechanisms of each pathway lays the foundation for further research on the selection and improvement of hypoxia-tolerant varieties of rice, broadens the mindset, and solves difficult problems.

## 2. Molecular Genetic Basis for Hypoxic Germination Tolerance in Rice

Mapping, cloning, and functional analysis of hypoxic germination tolerance genes and mining of hypoxia-tolerant rice genotypes are the basis for selection and breeding of direct-seeded rice varieties and further studies of the molecular genetic basis of hypoxia germination in rice.

### 2.1. Mapping of QTLs for Hypoxia Tolerance in Rice

In recent years, numerous QTLs and candidate genes for hypoxia germination tolerance have been mapped, and the application of genome-wide association study (GWAS) has facilitated more precise mapping of relevant QTLs and genes and paves a new way for their application in production [35,36]. Angaji et al. [37] used a backcross population with IR64 as a recurrent parent to be crossed with Khao Hlan On (KHO), and based on rice plant survival in a low-oxygen environment, five putative QTLs were detected on chromosome 1 (qAG-1-2), 3 (qAG-3-1), 7 (qAG-7-2), and 9 (qAG-9-1 and qAG-9-2), which could explain 17.9%–33.5% of the total phenotypic variations; colocalization of the QTLs was detected by genotyping. Later, Kretzschmar T. et al. [24] further identified the trehalose-6-phosphate phosphatase gene (OsTPP) derived from KHO as a genetic determinant in qAG-9-2 using a population of NILs. They found that *OsTPP7* is involved in trehalose-6-phosphate (T6P) metabolism, and its activity could indicate a low sugar utilization rate by increasing the rate of T6P metabolism, which would in turn enhance starch mobilization to drive the growth kinetics of germinating embryo and elongating coleoptile, thereby enhancing hypoxic germination tolerance.

In QTL mapping and GWAS, coleoptile elongation, hypoxic germination rate, and seedling survival are often used as indicators of hypoxic germination tolerance in rice. Moreover, qAG7.1 (31.7% contribution) was localized to chromosome 7 using a cross population of Ma-Zhan Red with IR42 [38]. Baltazar [39] constructed an F2:3 population with Nanhi and IR64 as the parents and utilized 300 genotypes. By SNP genotyping, a major QTL qAG7 from Nanhi was localized on chromosome 7 with LOD of 13.93–22.3%, and qAG11 on chromosome 11. GWAS and biparental QTL mapping revealed that *LOC_Os01g53930* located on chromosome 1 could explain 27% of the phenotypic variance and encodes hexokinase (HXK6), which potentially mediates allelic variation between indica and japonica varieties and is located within a 158 kb interval between indica and japonica, resulting in phenotypic variance in the response of coleoptiles to hypoxic conditions [40]. Sun K et al. [41] used 200 rice germplasms and detected significant changes in the expression of Os02g0657000, Os03g0592500, and Os08g0380100 on chromosomes 2, 3, and 8, showing sensitivity of these genes to oxygen treatment. Zhang M et al. [42] selected 432 indica rice varieties to construct a natural population, and detected two genes, *LOC_Os08g39290* and *LOC_Os06g03250*, which were both hypoxia-induced and highly expressed in flood-tolerant varieties, using the length of flooded coleoptile as an indicator. Further sequencing analysis of the candidate gene showed that *LOC_Os06g03250* is a highly hypoxia-inducible gene, which is very helpful for future research on the molecular mechanism of rice hypoxia tolerance.

In addition, some researchers [43] utilized 191 japonica rice genotypes from different regions and years to screen hypoxia-tolerant genotypes. A total of four QTLs (qGS1, qGS3, qGS9, and qGS10) were mapped by field simulation of flooding tolerance, located on chromosomes 1, 3, 9, and 10, respectively. Another study [44] utilized the DH population in a rice seedling flooding tolerance test and detected a total of 16 QTLs associated with seedling flooding tolerance. Among these QTLs, four are related to mesocotyl length, three to plant height (PH), three to survival, three to damage index of chlorophyll (DIOC), and three to relative damage percent of dry weight (RDP), with RDP = (control dry weight − flooded treatment dry weight) × 100%/control dry weight. These QTLs are located on chromosomes 1, 2, 3, 4, 6, 8, 9, and 12, and their contributions range from 10.7% to 41.4%. In another study, the length of coleoptile under hypoxic stress was used as an indicator, and two QTLs (qSAT-2-R and qSAT-7-R) associated with hypoxic germination tolerance in rice were mapped, which could explain 8.7% and 9.8% of the phenotypic variance, respectively [45]. Furthermore, a DH population constructed with the indica rice variety Taichung Native 1 (TN1) and the japonica rice variety Chun 06 (CJ06) as parents was used to examine the hypoxic germination tolerance of rice using seedling survival as an indicator. As a result, six QTLs were mapped on chromosomes 1, 2, 6, 8, and 9, in which the *LOC_Os08g42750* gene in the QTL interval on chromosome 8 is of great reference value for fine-tuning the hypoxia tolerance of rice and molecular breeding [46] (Table 1).

Miriam D and Baltazar used Kharsu 80A as a donor parent to be crossed with IR64, selected 190 phenotypic populations based on hypoxia tolerance, and identified four QTLs through GWAS, including three on chromosome 7 (qAG7.1, qAG7.2, and qAG7.3) and one on chromosome 3 (qAG3), with LOD values ranging from 5.7 to 7.7 and phenotypic variation explanation of 8.1–12.6% [47]. Ghosal S et al. [48] crossed the indica variety Kalarata with the recurrent parents NSIC Rc222 and NSIC Rc238, and performed QTL analysis, resulting in the identification of QTLs on chromosomes 3, 5, 6, 7, and 8 by survival rate (SUR) and chromosomes 1, 3, and 7 by seedling height (SH) using the BC1F2:3 population. Among the five QTLs for SUR, the second largest QTL (qSUR6-1) is novel for hypoxic germination potential in rice, which could explain 11.96–16.01% of the phenotypic variation. The QTL for SH (qSH1-1) is also novel, which could explain 13.53–34.30% of the phenotypic variation. Another study utilized 209 exceptional genotypes, and obtained four candidate genes (Os01g0911700 (OsVP1), Os05g0560900 (*OsGA2ox8*), Os05g0562200 (OsDi19-1), and Os06g0548200) by GWAS and resequencing based on the length and diameter of coleoptile in an anaerobic environment. These genes are involved in the process of GA and ABA signaling and cellular metabolism, which are more valuable for screening favorable direct-seeded rice germplasm [49]. In another study, nine BC_2_ populations from the 260 introgression line (IL) were used to obtain 124 significant materials. As a result, GWAS identified 59 QTLs for drought tolerance and 68 QTLs for flooding tolerance, and some pleiotropic loci were found to have both flooding and drought tolerance [50].

The Kinmaz × DV85 RIL population was utilized for QTL analysis of low-temperature and hypoxic germinability. Five QTLs for low-temperature germinability were detected at five markers on chromosomes 2, 6, 7, 11, and 12, including X67, X386, R1440, G1465, and X148, whose contribution rates ranged from 8.8% to 27.1%. Five QTLs for hypoxic germinability were detected on chromosomes l, 2, 5, and 7, and the two QTLs on chromosome 5 were located in the vicinity of the G260 and X105 markers, with contribution rates ranging from 12.05% to 19.56% [51].

At present, rice under hypoxia stress is mostly examined by indicators such as germinal sheath length, mesocotyl length, and seedling survival, and previous research used RIL, BIL, DH, NIL, BC_2_F_2_, and other populations for hypoxia-tolerant QTL localization and GWAS analysis [52]. More than 40 QTLs related to hypoxia germination tolerance in rice have been reported, which are widely distributed on 12 chromosomes [53]. Hypoxic germination tolerance is a complex trait co-regulated by the population and the environment. Moreover, in actual production, direct seeding of rice not only encounters hypoxia but also suffers from a variety of other stresses, such as low temperature and drought. Therefore, mining of a variety of genes for stress tolerance is crucial for the breeding of rice.

**Table 1 ijms-25-02177-t001:** QTLs related to hypoxia tolerance in rice.

Combinations of Parents	Group Type	Evaluation Index of Hypoxia Resistance	Range of Contribution Rates (R^2^)	QTL/Candidate Genes	Chromosomes	References
Khao Hlan On × IR64	BC_2_F_2_	Survival percentage	17.9–33.5%	*qAG1-2*, *qAG3-1*, *qAG7-2*, *qAG9-1*, *qAG9-2*	1, 3, 7, 9	[42]
Ma-Zhan Red × IR42	F_2:3_	Survival percentage	31.7%	*qAG7.1*	7	[38]
Nanhi × IR64	F_2:3_	Seedling survival rate	13.93–22.3%	*qAG7*, *qAG11*, *qAG2.1*	2, 7, 11	[39]
Nipponbare × IR64	RIL	Anaerobic response index	27%	*qAG1-2*	1	[40]
ASD1 × IR64	F_2:3_	Survival percentage	15.1–29.4%	*qAG7*, *qAG9*	7, 9	[54]
Lianjing 15 × Huanglizhan	F_2:3_	Germ sheath length	11.7–24%	*qGS1*, *qGS3*, *qGS9*, *qGS10*	1, 3, 9, 10	[43]
TN1 × CJ06	DH	Mesocotyl length, Chlorophyll damage index, plant height, survival rate, dry mass relative damage rate	10.6–41.1%	*qLOE-12* *qPH12* et al.	1, 2, 3, 4, 6, 8, et al.	[45]
Xiushui79 C Bao, NIP × Kasalath	RIL, BIL	Anoxic response index	5.8–16.2%	*qSAT-2-R*, *qSAT-2-B* et al.	2, 3, 5, 7, et al.	[55]
Kinmaz × DV85	RIL	Anaerobic germination	12.05–19.06%	*qAG-1*, *qAG-2* et al.	1, 2, 5, 7, et al.	[51]
USSR5 × N22	F_2:3_	Anoxia germinability	10.99–15.51%	*qAG-5*, *qAG-11*	5, 11	[54]
Kharsu80A × IR64	F_2:3_	Seedling survival rate	8.1–12.6%	*qAG7.1*, *qAG7.2*, *qAG7.3*, *qAG3*	3, 7	[47]
94 rice genotypes	-	Anaerobic germination	>20%	*LOC_Os03g31550*, *LOC_Os12g31350*	3, 12	
432 Indian rice	-	Germ sheath length	-	*LOC_Os06g03520*	6	[56]
Kalarata × NSIC Rc222/NSIC Rc238	BC_1_F_2:3_	Survival, seedling height	11.96–16.01%, 13.53–34.30%	*qSUR3-1*, *qSUR5-1*, *qSUR6-1*, *qSH1–1* et al.	1, 3, 5, 6, et al.	[48]
Zhaxima × Nanjing46	RIL	Coleoptile length	11.24%	*qAG-12*	12	[52]
209 natural rice populations	-	Coleoptile length (CL) and coleoptile diameter (CD)	-	Os01g0911700, Os05g0560900, Os05g0562200, Os06g0548200	1, 5, 6	[49]

### 2.2. Phytohormone Regulation and Sugar Metabolism Pathways in Hypoxia-Tolerant Rice

Rice regulates sugar metabolic processes and phytohormonal pathways in response to hypoxic stress. Sugar metabolism is a respiration-related metabolic process. Under hypoxic conditions, aerobic respiration gradually decreases and anaerobic respiration increases [57]. The ethanol fermentation pathway with anaerobic respiration is the most prevalent, which is mainly due to the hypoxia-induced elevation of pyruvate decarboxylase, ethanol dehydrogenase, and acetaldehyde dehydrogenase activities in the alcoholic fermentation pathway [58]. Of all studied cereals such as wheat, corn, and soybean, only rice has the ability to rapidly elongate its coleoptile in a hypoxic environment [59]. Among these enzymes, α-amylase is one of the key enzymes in glycolysis, and hypoxia-tolerant varieties generally show higher α-amylase activity than susceptible varieties under hypoxic stress [60]. It was found that Ideal Plant Architecture 1 (IPA1) ipa1-NIL seeds had decreases in starch metabolism, as indicated by analysis of soluble sugar content, amylase activity, and α-amylase, whereas GA inactivation *OsGA20ox1* genes were up-regulated, suggesting that IPA1 can regulate starch metabolism through GA [61]. The metabolic strength of starch, the main energy substance for rice seed germination, directly determines the ability of rice to tolerate hypoxic stress. Tricarboxylic acid cycle and oxidative phosphorylation are inhibited under anaerobic conditions [62,63]. A recent report found that the aquaporin NIP2 mediates lactate transport [64]. The subfamily of aquaporins may have some important lactate specific functions in plants under hypoxic conditions because NIP2 mediates lactate transport and causes anaerobic respiratory ethanol fermentation [65].

The auxin is indole acetic acid (IAA), and the rapid elongation of coleoptiles under hypoxic stress is often closely related to auxin. Under hypoxic stress, miR393 expression is suppressed to promote the elongation and growth of the radicle sheath, in which OsMIR393b is expressed in shoot meristematic tissues, radicle sheaths, and stomatal cells [66]. Under hypoxia, ethylene can also promote coleoptile growth by repressing the expression of GY1 and other genes in JA biosynthesis [67]. For rice under flooding, a rapid decrease in ABA concentration is a prerequisite for accelerated stem elongation. The mRNA level of *OsABA8ox1* was extremely significantly elevated at 1 h after flooding, and the expression of *OsABA8ox1* was rapidly induced in seedlings grown under normal conditions by treatment with ethylene and its precursor 1-aminocyclopropane-1-carboxylic acid (ACC) [68]. Rice flooding increases the content of active GA(1) and its immediate precursors by enhancing the expression of gibberellin GA biosynthesis genes (*OsGA1ox53* and *OsGA1ox19* and *OsGA20ox20*). Alterations of GA metabolism would increase the content of GA(1) and subsequently enhance the rice coleoptile elongation rate [69]. Transcriptome analysis of hormones revealed that polybutazole (PB) improves rice survival under flooding by maintaining photosynthesis and reducing nutrient metabolism [70]. Recently, it was found that ethylene plays a crucial role not only in response to flooding but also in metabolic acclimatization of plants, and improves rice survival at the seedling stage by inducing specific transcription factors [71].

### 2.3. Hypoxia Tolerance Genes and Functions in Rice

Breeding for hypoxia tolerance in rice relies on the construction of high-density molecular marker linkage maps and identification of hypoxia-tolerant mutants to map the desired genes. It has been demonstrated that hypoxia tolerance in rice is mainly regulated by genetic factors.

We used the National Rice Data Center (https://ricedata.cn/gene/ (accessed on 26 September 2022)) to screen rice-related genes for the keywords “flooding tolerance, submergence tolerance, seedling germination rate” as well as factors and pathways involved in the biological processes of rice genes in response to the low-oxygen environment, then analyzed the screened genes according to their names and biological functions and classified them into five categories (Table 2). The *OsTPP7* gene has been successfully cloned, which may reduce sugar utilization by increasing the turnover of trehalose-6-phosphate to enhance the reservoir capacity in heterotrophic tissues, thereby enhancing starch utilization, promoting the growth of germinating embryos and coleoptiles, and improving the tolerance of rice to hypoxic germination [24]. *OsCIPK15* encodes a protein kinase that promotes rice germination under hypoxic conditions by phosphorylating the stress receptor SnRK1A and activating the SnRK1A-MYBS1-mediated sugar signaling pathway to promote saccharide catabolism [72]. *OsABF1* is an auxin binding factor participating in rice hypoxia stress response by promoting the aboveground and root growth under hypoxia stress [26]. Hypoxia induces the expression of the *OsHIGD2* gene, which in turn promotes the growth of rice stems under hypoxic stress [25]. The *Adh1* gene encodes a rice ethanol dehydrogenase that reduces acetaldehyde to ethanol during alcoholic fermentation. It is required for the synthesis of ATP by plants under anaerobic conditions for alcoholic fermentation, and promotes the elongation of rice coleoptiles under flooded conditions [28].

Previously, indica cultivar FR13A was localized to a chromosome 9 QTL, which is called *Sub1* and has three genes, *Sub1A*, *Sub1B*, and *Sub1C*, encoding putative ethylene response factors [73]. *Sub1A* is a rice flooding tolerance gene, whose expression is induced by ethylene treatment but not by GA treatment. Flood-tolerant rice varieties containing the *Sub1A* gene synthesize ethylene under flooded conditions, which promotes the degradation of ABA and the expression of *Sub1A*. *Sub1A* promotes the accumulation of the inhibitory factors SLR1 and SLRL1 in GA signaling and thus inhibits the GA response, which inhibits the elongation of the aboveground part of the plant under water and reduces carbohydrate consumption, ultimately increasing flooding tolerance [74]. A comparison of the transcriptome profiles of M202 varieties with Sub1A-1 and their relative absence of sub1A in response to flooding revealed that the flooding tolerance pathway is related to anaerobic respiration, phytohormone response, and antioxidant system, with the transcriptional family of AP2/ERF proteins, which is related to Sub1A-1-mediated flooding tolerance response. Overexpression of *SUB1A* enhanced the response to ABA, which activated the expression of adversity-induced genes [33]. SUB1A limits the accumulation of reactive oxygen species (ROS) in aboveground tissues during drought and release from flooding. Similarly, *SUB1A* increases the expression abundance of ROS scavenging enzymes, which in turn enhances tolerance to oxidative stress. Thus, in addition to enhancing rice tolerance to flooding stress, *SUB1A* increases rice survival during water deficit and rapid dehydration triggered by flooding release [75].

The *CIPK15* pathway is repressed in FR13A, a flooding-tolerant variety that carries the Sub1A gene, in a hypoxic environment. For the flooding-intolerant variety that elongates rapidly after flooding, *CIPK15* may be involved in up-regulating Ramy3D expression, whereas *Sub1A* does not have such function [25]. *SUB1A* genotypic plants alter the response to persistent darkness by limiting ethylene production and jasmonic and salicylic acid responses, preventing the degradation of chlorophyll and carbohydrates as well as accumulation of senescence-associated mRNAs. SUB1A-conferred delay in leaf senescence contributes to enhanced tolerance to flooding, drought, and oxidative stresses in rice [76].

MPK3 can interact with the flooding-tolerant allele *SUB1A* and phosphorylate *SUB1A*. In addition, under flooding stress, SUB1A1 can bind to the MPK3 promoter and regulate the expression of MPK3 through a positive regulatory loop [31].

**Table 2 ijms-25-02177-t002:** Genes related to hypoxia tolerance in rice.

Type	Genetic Symbol	Gene Annotation	MSU-Locus	References
Transcription factor	*OsABF1*; *OsbZIP12*	bZIP transcription factor	*LOC_Os01g64730*	[26]
	*OsPHR2*	MYB. transcription factor	*LOC_Os07g25710*	[77]
	*OsEREBP1*	EREBP. transcription factor	*LOC_Os02g54160*	[78]
	*OsbZIP72*	bZIP. transcription factor	*LOC_Os09g28310*	[79]
Gene	*Sub1A*	flood-tolerant gene	*LOC_Os09g28180*	[33]
	*OsCIPK15*	calcineurin-like neurophosphatase B subunit-interacting protein kinase gene	*LOC_Os11g02240*	[72]
	*OsNAAT1*	nicotinamide aminotransferase gene	*LOC_Os02g20360*	[80]
	*OsHIGD2*	hypoxia-induced gene	*LOC_Os07g47670*	[25]
	*MHZ6*; *OsEIL1*	mao huzi 6 gene	*LOC_Os03g20790*	[81]
	*OsABA8ox1*	ABA 8′-Hydroxylase gene	*LOC_Os02g47470*	[82]
	*Adh1*	alcohol dehydrogenase gene	*LOC_Os11g10480*	[28]
	*OVP3*	vacuolar H+-pyrophosphatase gene	*LOC_Os02g55890*	[83]
	*OsAmy3D*	alpha-amylase isozyme 3D	*LOC_Os08g36910*	[84]
	*D14L*	DWARF 14 LIKE; alpha/ beta-fold hydrolase	*LOC_Os03g32270*	[85]
	*OsSMAX1*	SMAX1-Like (SMXL) gene	*LOC_Os08g15230*	[86]
	*OsABA8ox3*	ABA 8′-Hydroxylase	*LOC_Os09g28390*	[87]
	*OsETOL1*	homolog of ETHYLENE OVERPRODUCER	*LOC_Os03g18360*	[88]
	*OsACS2*	aminocyclopropane-1-carboxylate synthase	*LOC_Os04g48850*	[89]
	*OsiSAP8*	O. sativa subspecies indica stress-associated protein gene	*LOC_Os06g41010*	[90]
	*OsTPP7*	trehalose-6-phosphate phosphatase	*LOC_Os09g20390*	[24]
Kinase	*SnRK1A*	Snf1 protein kinase	*LOC_Os05g45420*	[91]
	*OsMAP1*	Mitogen-activated protein	*LOC_Os03g17700*	[92]
	*OsGSK2*	GSK3/SHAGGY-like kinase	*LOC_Os05g11730*	[93]
	*OsPAO5*	polyamine oxidase 5	*LOC_Os04g57560*	[94]
	*OsPME1*	pectin methyl esterase	*LOC_Os03g19610*	[95]
	*OsCRTISO*	carotenoid isomerase gene; zebra leaf	*LOC_Os11g36440*	[96]
	*OsBADH1*	betaine aldehyde dehydrogenase	*LOC_Os07g48950*	[97]
	*OsCCD7*; *HTD1*	carotenoid cleavage dioxygenase	*LOC_Os04g46470*	[98]
Responsive factor	*OsEIL1*; *OsEIL1a*	ethylene-insensitive	*LOC_Os03g20780*	[99]
Protein	*CycP2;1; CYC U2*	U-type cyclin	*LOC_Os04g46660*	[100]
	*SLR1*; *OsGAI*; *Slr1-d*	slender rice 1; GRAS-domain protein	*LOC_Os03g49990*	[101]
	*OSISAP1*; *OsSAP1*	O. sativa subspecies indica stress-associated protein gene	*LOC_Os09g31200*	[102]
	*OsDOG*; *OsSAP11*	A20/AN1 zinc-finger protein	*LOC_Os08g39450*	[103]

## 3. Sequence Variations and Hormonal Responses of Rice Hypoxia-Tolerant Genes

### 3.1. Formatting of Mathematical Components

Plants contain a myriad of metabolic pathways responsible for the biosynthesis of complex metabolites, and rice hypoxia tolerance genes use metabolic pathways such as glycolysis, the tricarboxylic acid cycle, and ethanolic acid oxidation. When rice is under flooding, it transmits hypoxia messages to hypoxia-tolerant genes to regulate the formation of sugar and the expression of ethanol dehydrogenase, which serves as a substrate for glycolysis, and the subsequently formed pyruvic acid is oxidized by ethanol dehydrogenase to form ATP, providing energy for the growth of rice and improving the tolerance of rice to flooding [25].

To further investigate the role of hypoxia tolerance genes in plant metabolic pathways and to understand the biological functions of genes, 33 rice hypoxia tolerance-related genes were searched in the national rice database with KEGG analysis (Kyoto Encyclopedia of Genes and Genomes (KEGG) databases) (https://www.kegg.jp/kegg, (accessed on 26 September 2022)). KEGG is a major public pathway-related database that provides classifications valuable for studying genetics and biologically complex behaviors. It contains systematic analyses of the intracellular metabolic pathways and functions of gene products, which can help to study the complex biological behavior of genes. There are 33 genes related to hypoxia tolerance in rice, most of which are associated with secondary metabolite biosynthesis, endoplasmic reticulum protein processing, and phytohormone signaling [104]. The KEGG analysis results demonstrated three hypoxia stress-related genes in the MAPK signaling pathway, eighteen hypoxia stress-related genes in the secondary metabolite biosynthesis pathway, four hypoxia stress-related genes in the carotenoid biosynthesis pathway, and five hypoxia stress-related genes in the plant hormone signal transduction (Figure 1).

Among them, *OsMAP1*, *OsEIL1*, *OsEIL1a*, and *MHZ6* are mainly enriched in the osa04016 (MAPK signaling pathway) pathway with three levels of signaling process, MAPK, MEK, and MEKK. These three kinases are sequentially activated and together regulate plant growth and development and phytohormone accumulation. *OsMAP1* is located on the MAPK cascade reaction, and can interact with *OsMEK1* (MEK) and participate in the signaling pathway of low-temperature stress response in rice. *MHZ6* is present not only in the MAPK signaling pathway, but also in osa04075 (plant hormone signal transduction). The osa04075 pathway mainly consists of auxin (AUX), abscisic acid (ABA), and salicylic acid (SA)-related pathways, with ABA playing a key role in hypoxic stress response. *MHZ6* is in the ethylene signaling pathway, and promotes internode elongation in deep-water rice by activating the *SD1* gene [105]. *OsAmy3D*, *Adh1*, and *OsTPP7* are mainly enriched in the osa01110 (biosynthesis of secondary metabolites) pathway, a process by which organisms synthesize non-essential substances for life and store secondary metabolites in the plant; this process is very complex, where *OsAmy3D* and *Adh1* have similar indica–japonica varietal differentiation. Understanding the distribution and role of rice hypoxia tolerance genes in metabolic pathways can lay a foundation for further research on the application of hypoxia tolerance genes in genomics research.

It has been demonstrated that hypoxia tolerance genes are affected by phytohormone signaling. To investigate the response of hypoxia tolerance genes to phytohormones, rice hypoxia genes were subjected to phytohormone response expression profiling by the Rice Expression Profiling Database (http://ricexpro.dna.affrc.go.jp/ (accessed on 28 September 2022)) [106] (Figure 2). Under the treatment of ABA, the expression of Os09g0457100, Os01g0867300, Os02g0703600, Os11g0210300, Os07g0673900, and Os09g0486500 was continuously up-regulated in roots and aboveground over time; that of Os02g0802500 was up-regulated in stems and down-regulated in roots for the first 6 h; that of Os09g0454900 was first up-regulated in stems, then remained unchanged from 1 to 3 h, and showed no significant change in roots. Under ABA treatment, Os03g0294700, Os02g0802500, Os07g0673900, Os04g0578000, Os04g0550600, and Os04g057800 were up-regulated in the stem, but showed no significant change in expres/sion in roots. Under cytokinin (CTK) treatment, the expression of Os03g0285800 was first up-regulated, then down-regulated in roots. Os02g0802500 and Os09g0306400 were down-regulated in roots, but their expression showed no significant change in stems; the expression of Os011g0113700 was significantly down-regulated in stems at 6–12 h and there was no significant change in roots. Under jasmonic acid (JA) treatment, Os03g0285800, Os04g0550600, and Os07g0673900 were up-regulated in roots, but showed no significant change in expression in stems, whereas Os04g0578000 was up-regulated in both roots and stems under ABA, JA, and CTK treatments. *OsCIPK15* was significantly down-regulated in stem and unchanged in roots under CTK and JA treatments. Under ABA treatment, *OsCIPK15* showed no significant change in roots, while its expression in stems was down-regulated in the first 0.5 h and up-regulated after 12 h. Moreover, the expression of *OsCIPK15* in stems was significantly down-regulated under CTK and JA treatment.

The above results reveal that hypoxia tolerance genes are up- or down-regulated to different degrees in response to ABA, CTK, and JA treatments, whereas their expression shows no significant response to gibberellins (GAs) and brassinosteroids (BRs). It is well-known that genes for phytohormone metabolism and signal transduction are related to important agronomic traits that determine crop yield. However, there has been no report on the expression profiling of hypoxia tolerance genes in response to phytohormone metabolism and signal transduction in rice. Analysis of the expression profiles of hypoxia-tolerant genes in response to various phytohormones can characterize the expression profiles of rice hypoxia genes [106].

### 3.2. Analysis of Population Differences in Genes

Presence/absence variation (PAV) analysis of 33 hypoxia tolerance genes was performed based on the 3K Asian cultivated rice pan-genome database (https://www.rmbreeding.cn/Genotype/haplotype, (accessed on 18 September 2023)). The results showed that hypoxia tolerance genes could be categorized into three groups based on the frequency of genes in different subspecies and subpopulation clusters, including core genes, candidate core genes, and distributed genes [107]. There are twenty-four core genes, one candidate core gene Os07g0689150, and five distributed genes Os01g0867300, Os02g0306401, Os08g0250900, Os09g0306400, and Os09g0369400. The genes with higher frequencies in certain subspecies than in other subspecies are called “subspecies imbalance genes”, which are further categorized into two subcategories: indica-dominant genes, whose frequency in indica is 5% or above higher than that in japonica, and japonica-dominant genes, whose frequency in japonica is 5% or above higher than that in indica. The frequency of Os09g0306400 was 83% in indica rice and 98.8% in japonica rice, suggesting that this gene is a japonica-dominant gene; while Os01g0867300 was distributed in 92.2% of indica rice, suggesting that it is an indica-dominant gene. The frequency of Os09g0369400 was 59.9% in indica rice and 53.8% in japonica rice, indicating that this gene is an indica-dominant gene. In addition, Os02g0306401 and Os08g0250900 were more evenly distributed in rice subgroups as random genes. The dominant alleles of different geographically local populations can indicate a strong correlation between the ancestral genotype and wild rice material from the same geographical origin [108] (Figure 3).

### 3.3. Haplotype Analysis of Hypoxia Tolerance Genes

To further explore the use of hypoxia tolerance genes in breeding, the 3K Asian cultivated rice pan-genome database (https://www.rmbreeding.cn/Genotype/haplotype, (accessed on 18 September 2023)) was utilized, and the hypoxia tolerance genes were analyzed by gene-coding sequence haplotype (gcHap). Combined with the results of public databases, 33 hypoxia resistance genes were analyzed by gcHap, some of which could not be mapped due to base deletion or other reasons [108].

In the gcHap analysis of known rice hypoxia tolerance genes, Figure 4 shows the presence of 3–5 network haplotypes for most of the genes based on the number of samples n > 50, such as *OsPHR2*, *MH26*, *OsHIGD2*, *OVP3*, and *OsAm3D*. *OsPHR2*, a MYB structural domain-containing transcription factor with partially redundant functions, is expressed in cortical and mesophyll cells of the primary and lateral roots as well as in the chloroplasts of the leaves of rice [77]. *OsPHR2* has three major haplotypes (Haps), with the highest percentage of Xian in Hap1, which is almost exclusively present in indica rice, and a single nonsynonymous mutation between Hap2 and Hap3. *Sub1A* is a rice flooding tolerance gene, whose expression is up-regulated under flooding conditions and induced by ethylene treatment but not by GA treatment. It encodes three ethylene response factors (ERFs), and rice carrying the Sub1A-1 allele is more tolerant to flooding [33]. *Sub1A* has two major Haps and three minor Haps, with Hap1 having a high percentage of Xian, Geng, and Aus, and Hap3 being dominated by Xian and Geng. *OsCIPK15* encodes a calmodulin-like neurophosphatase B-subunit-interacting protein kinase that regulates carbohydrate catabolism and fermentation through the SnRK1A-MYBS1-mediated sugar signaling pathway, and regulates sugar and energy production and enables rice growth under flooded conditions and germination under hypoxic conditions [72]. *OsCIPK15* has three major Haps, among which Hap2 is the highest in Xian and there are two non-synonymous mutations between Hap1 and Hap2. Hap3 has a high percentage of Xian and Geng. *OsTPP7* increases the sugar utilization by increasing turnover of alginate-6-phosphate, enhances starch utilization, drives the growth of germinating embryos, lengthens the coleoptile, and improves tolerance to hypoxic germination [24]. *OsTPP7* has three major Haps, including Hap1 (n = 1109), Hap2 (n = 1349), and Hap3 (n = 217). Hap1 and Hap3 have the highest frequency in Xian and are mainly found in the indica subgroups, whereas Hap1 differs from Hap3 by a single nonsynonymous mutation. Hap2 has a high percentage of Geng and is mainly found in the genetic background of japonica rice. The results of haplotype network analysis of this gene are consistent with those of PAV analysis.

*OsHIGD2* is a hypoxia-inducible gene with high expression levels in seeds, mesocotyls, roots, and mature leaves, and can interact with proteins involved in early signaling of hypoxia-promoted stem growth in deep-water rice. Its expression is also increased by ethylene [24]. *OsHIGD2* has two major Haps, with a high percentage of Xian and Geng in Hap1 and a high percentage of Geng in Hap2, which is mainly dominated by japonica, and Hap1 and Hap2 are distinguished by a single nonsynonymous mutation. *MHZ6* encodes *OsEIL1* to act downstream of *OsEIN2*, and positively regulates the ethylene response of rice roots; *MHZ6* is expressed in both roots and coleoptiles of seedlings, and its overexpression increases seed size and TGW [34]. *MHZ6* has three major Haps and one minor Hap. Hap1 has a high percentage of Geng, which is mainly present in japonica varieties, Hap2 has a high frequency of Xian and Geng dominated by japonica, and Hap3 has a high percentage of Xian dominated by indica genetic background. It is a typical gene for indica–japonica divergence. *OsMAP1* is a mitogen-activated protein kinase located in the same locus as *OsMPK3*, and *OsMEK1* and *OsMAP1* interact with each other and are expressed in the signaling pathway at low temperature (12 °C). *OsMPK3* interacts with and phosphorylates the flooding-tolerant allele *SUB1A* and under flooding stress, and *SUB1A* binds to the MPK3 promoter to regulate its expression through a positive regulatory loop [92]. *OsMAP1* has three major Haps. Hap1 and Hap2 have the highest frequency in Xian dominated by indica rice, and Hap1 is distinguished from Hap2 by a single non-synonymous mutation. Hap3 has the highest frequency of Geng dominated by japonica rice genetic background. *OsEREBP1* is a transcription factor of the AP2/ERF family. *OsEREBP1* overexpression activates jasmonate and abscisic acid synthesis pathway-related genes as well as defense response signals to improve drought and flooding tolerance in transgenic rice [78]. *OsEREBP1* has four major Haps. Hap2 has high frequencies of Xian, Geng, Bas, and Adm, which are mainly dominated by japonica. Hap3 has high frequencies of Xian, Aus, and Adm, which are mainly dominated by indica. Adh1 encodes a rice ethanol dehydrogenase required for ATP synthesis by plants under anaerobic conditions for alcoholic fermentation, and promotes the elongation of rice coleoptiles under flooded conditions [28]. Adh1 has three major Haps and two minor Haps. Hap1 has the highest percentage of Geng dominated by japonica. Hap2 has the highest frequency of Xian, and there is a single non-synonymous mutation between Hap2 and Hap3. Hap5 has the highest percentage of Xian and Geng, which is mainly dominated by japonica. *OVP3*, which encodes vesicular H^+^-pyrophosphatase, is induced by hypoxia, particularly in the coleoptiles of flood-tolerant rice varieties, but is not induced by salt and cold stresses, suggesting that *OVP3* may play a role in the growth of rice under hypoxic conditions [83]. *OVP3* has two major Haps and three minor Haps. Hap1 has the highest percentage of Geng dominated by japonica, and Hap5 has a high frequency of Xian, which is almost exclusively present in indica; the difference between Hap1 and Hap5 lies in a single non-synonymous mutation. *OsAmy3D* is an α-amylase inhibiting oxidative phosphorylation through hypoxia, and interferes with the repression of sugar on the Amy3 gene subfamily, thereby allowing up-regulation of the expression of members in this family during hypoxic germination of rice embryos [84]. *OsAmy3D* has three major Haps. Hap1 has a high frequency in Geng and is almost exclusively present in japonica; Hap2 has a high percentage in Xian and is present mainly in indica. Therefore, *OsAmy3D* is a typical gene for indica–japonica divergence. Hap3 has a high frequency of Xian, Bas, Aus, and Adm dominated by indica. Moreover, there is a single non-synonymous mutation between Hap2 and Hap3.

The above gcHap analysis reveals obvious indica–japonica differentiation and that most of rice hypoxia tolerance genes are distributed in indica and japonica varieties. The results indicate that the genetic background of the hypoxia tolerance genes is mainly dominated by indica and japonica rice, and it is of great significance to further explore more resistance mechanisms of the hypoxia tolerance genes to improve the hypoxia tolerance of rice by various means, such as artificial selection and natural domestication.

## 4. Discussion and Prospects

### 4.1. Discussion

This review summarizes the genes and regulatory mechanisms of hypoxic tolerance germination of rice under hypoxia stress from the perspectives of physiology, biochemistry, molecular genetics, sequence variations, and hormone responses. In addition, we also reviewed the QTLs mapped by molecular marker-assisted selection (MAS), GWAS fine mapping of hypoxia tolerance genes [109], and categorization and functional analysis of 33 hypoxia tolerance genes. Analysis of gene sequence variation and hormone response yielded three pathways in hypoxia tolerance genes. One of them, *OsMAP1* is located on the MAPK cascade reaction of the osa04016 pathway. *MHZ6* is present in both the MAPK signaling pathway and the osa04075 pathway. In the gene haplotype network analysis of *MHZ6*, there is a high percentage of Geng in Hap1, which mainly exists in japonica varieties, while Hap3 is mainly dominated by indica varieties, which is a clear indica–japonica differentiation. *MHZ6* is involved in the ethylene signaling pathway, and promotes internode elongation in deep-water rice by activating the *SD1* gene [105]. *OsAmy3D*, *Adh1*, and *OsTPP7* are mainly enriched in the osa01110 pathway, where *OsAmy3D* and *Adh1* have similar indica–japonica differentiation and are mainly distributed in indica rice varieties. The frequency of Hap2 of *OsTPP7* was higher in the japonica rice subpopulation. The genetic background of this gene was derived from the japonica rice subpopulation.

*Sub1* has been shown to be a rice flooding tolerance gene with significantly up-regulated expression under flooding conditions [33], and it can encode members of the ethylene response factor group VII gene family, *Sub1A*, *Sub1B*, and *Sub1C*, and the expression of *Sub1A* is induced by ethylene signaling, which enhances flooding tolerance in rice by repressing GA-induced carbohydrates [110]. The action mechanism of *Sub1B* and *Sub1C* in *Arabidopsis* has been clarified, but how these two genes induce ethylene transcription factors in rice and the specific mechanism remain to be determined [111].

In further studies, through genotypic and phenotypic screening, we introduced AG1 into the excellent variety Ciherang-Sub1 using IR64-AG1 as the donor parent, which significantly improved the emergence tolerance of Ciherang-Sub1. A comparison of the genetic effects of AG1 and AG2 revealed that both of them not only significantly improve the emergence tolerance of rice, but also have a yield-enhancing effect, and at the same time have no negative effect on rice emergence and growth and development [112]. In breeding, superior germplasm resources carrying QTLs or flooding tolerance genes related to emergence tolerance can be utilized for hybridization. For instance, AG1 and the phosphorus-efficient gene Pup1 can synergize efficiently and be polymerized to improve both emergence tolerance and phosphorus uptake efficiently, and the polymerized line shows strong vigor and increased tillering at an early stage [113]. To address the problem of complete inundation due to flash floods in major watersheds, the Sub1 gene has been transferred into ten very popular locally adapted rice varieties, namely ADT 39, ADT 46, Bahadur, HUR 105, MTU 1075, Pooja, Pratikshya, Rajendra Mahsuri, Ranjit, and Sarjoo 52 [114].

*OsGF14h*, an upstream switch of ABA signaling in japonica-type weedy rice, inhibits the activity of the ABA receptor *OsPYL5* by interacting with two transcription factors, *OsHOX3* and *OsVP1*, in a hypoxic environment, reducing ABA sensitivity while activating GA biosynthesis, and ultimately inducing strong germination and emergence of weedy rice seeds. Further population genomics and evolutionary studies have shown that *OsGF14h* plays a key role in population differentiation between weedy and cultivated rice, as well as serves as a target gene for the genetic improvement of modern japonica rice, which provides a new research perspective and rationale for the domestication of cultivated rice in Asia [115]. The specific molecular mechanisms by which phytohormones regulate rice tolerance to hypoxic environments remain unclear. In addition, it remains unclear which pathways are involved in the synergistic interactions between various phytohormones under hypoxic stress, and the linkage between signaling and phenotypic variation under transcriptional regulation remains to be explored, which can help develop new rice breeding targets for direct seeding through genotypic analysis [116]. Various subgroups are included in the 3K database for their phenotypes under stresses such as salt, drought, and alkali tolerance, whereas there is little research on how the mechanisms underlying hypoxia tolerance are regulated in rice under hypoxic stress [117]. Therefore, it remains a great challenge to study the application of favorable haplotypes for hypoxia tolerance, population differences, and indica–japonica differentiation in rice breeding. Combined with molecular marker-assisted selection and background purity testing for selection and breeding, new rice varieties with better integrated agronomic traits and stronger flooding tolerance can be obtained after multiple generations of backcrossing and self-crossing. Moreover, the known germination flooding tolerance genes can be polymerized with other important genes for the breeding of direct-seeded rice varieties, which can provide more effective and practical breeding strategies for direct-seeded rice varieties under low-oxygen stress [118].

### 4.2. Prospects of Rice Hypoxia Tolerance Genes in Breeding

MAS and fine mapping of QTLs or cloned genes have been utilized for molecular breeding, and QTL mapping has been mostly carried out with the DH, RIL, and BIL populations with elite agronomic traits as the recurrent parents. However, the results of QTL mapping can hardly be directly applied to breeding populations [119]. Secondly, QTL localization is controlled by pleiotropic genes, and some minor-effect genes for hypoxia tolerance cannot be mined due to technical limitations. GWAS may become a trend of development because it can overcome QTL deficiencies. Although GWAS may have deficiencies such as false positives, with the development of bioinformatics, epigenetics, transcriptomics, genomics, and metabolomics, these deficiencies are expected to be overcome, and favorable, excellent, and highly resistant genes can be applied to rice breeding to further improve direct seeding of rice [120,121]. Hypoxia tolerance in rice is affected by a variety of intrinsic and extrinsic factors. Sugar metabolism plays a signaling role in hypoxia resistance of rice. EXPA7 and EXPB12 of the EXPA family have been reported to elongate the coleoptiles under hypoxia, and their expression is down-regulated under the action of the enzyme in hypoxia, suggesting that the genes have an energy-saving strategy [77]. Understanding the response mechanism of rice to hypoxia, in-depth mining of hypoxia tolerance genes, and screening of hypoxia-tolerant rice genotypes will be important for the breeding of hypoxia-tolerant rice varieties and the development of direct seeding of rice.

The MAS breeding method has been widely used in the selection of direct-seeded rice varieties under low-oxygen stress in rice. In addition, the gene editing technology CRISPR/Cas9 system can make genes related to biotic and abiotic stresses more functional through base mutations, deletions, or insertions [109]. For instance, the hypoxia genes *OsTPP7*, *Sub1A*, and *SD1-DW* were edited and cloned to obtain more hypoxia-tolerant varieties. Molecular markers, gene editing technology, transcriptome analysis, and modern breeding technology will be integrated to solve major problems in direct seeding of rice under low-oxygen stress that affect seed germination, seedling emergence, fruiting rate, and yield.

## Figures and Tables

**Figure 1 ijms-25-02177-f001:**
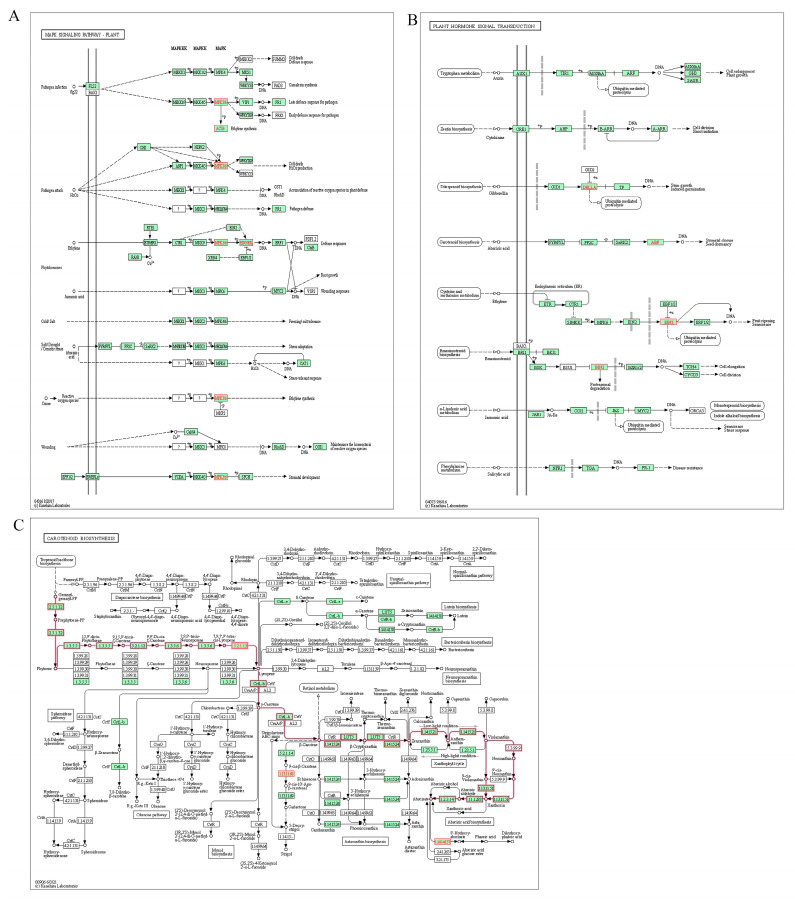
KEGG pathway analysis of (**A**) osa04016 (MAPK signaling pathway), (**B**) osa04075 (plant hormone signal transduction), and (**C**) osa00906 (carotenoid biosynthesis).

**Figure 2 ijms-25-02177-f002:**
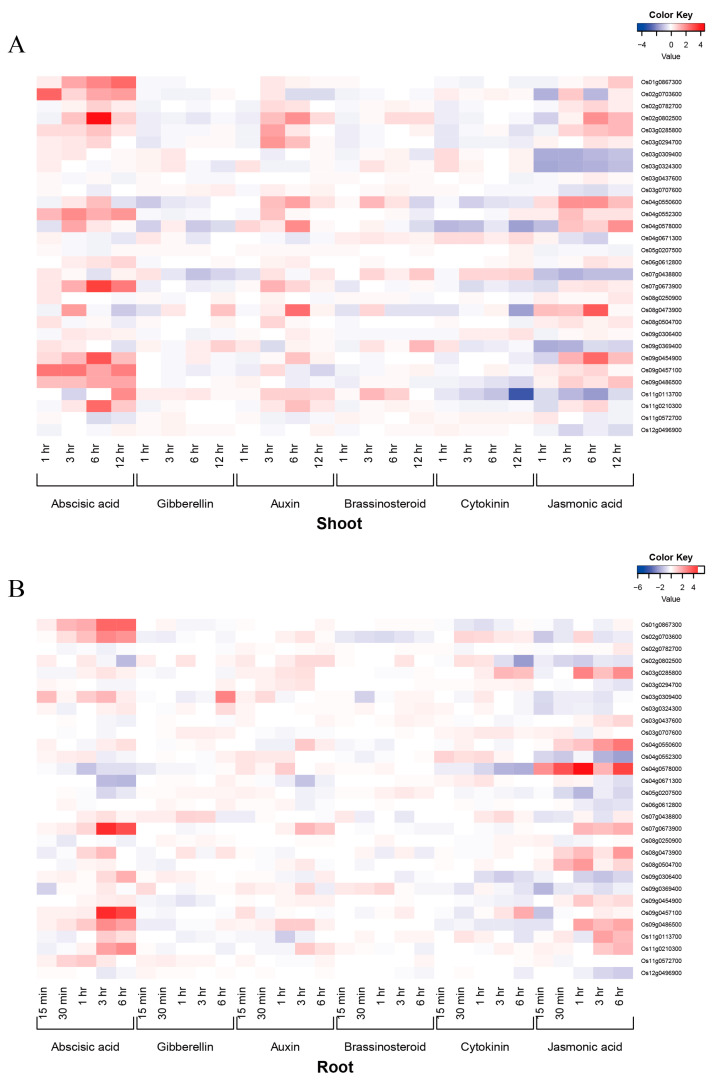
Heat map for the response of hypoxia tolerance genes to hormones in roots and shoots of rice. A: the response of hypoxia tolerance genes to hormones in shoots; B the response of hypoxia tolerance genes to hormones in roots. Hypoxia tolerance genes were analyzed by gene ID in the RiceXPro database to construct heat maps. The heatmap is constructed based on clustering of correlation distance and complete linkage of each gene using heatmap.2 in the “gplots” package of R program. The expression level for each gene is normalized by shifting the baseline of median value to zero for each gene across all data within a dataset.

**Figure 3 ijms-25-02177-f003:**
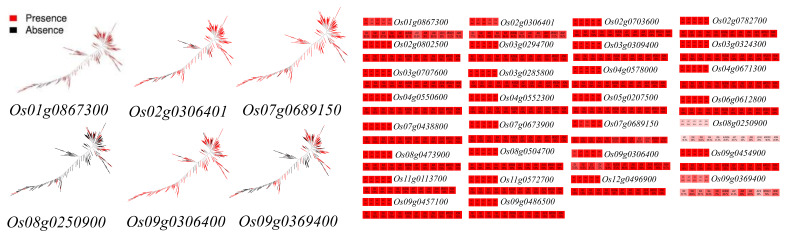
Distribution of hypoxia tolerance genes in different rice subgroups.

**Figure 4 ijms-25-02177-f004:**
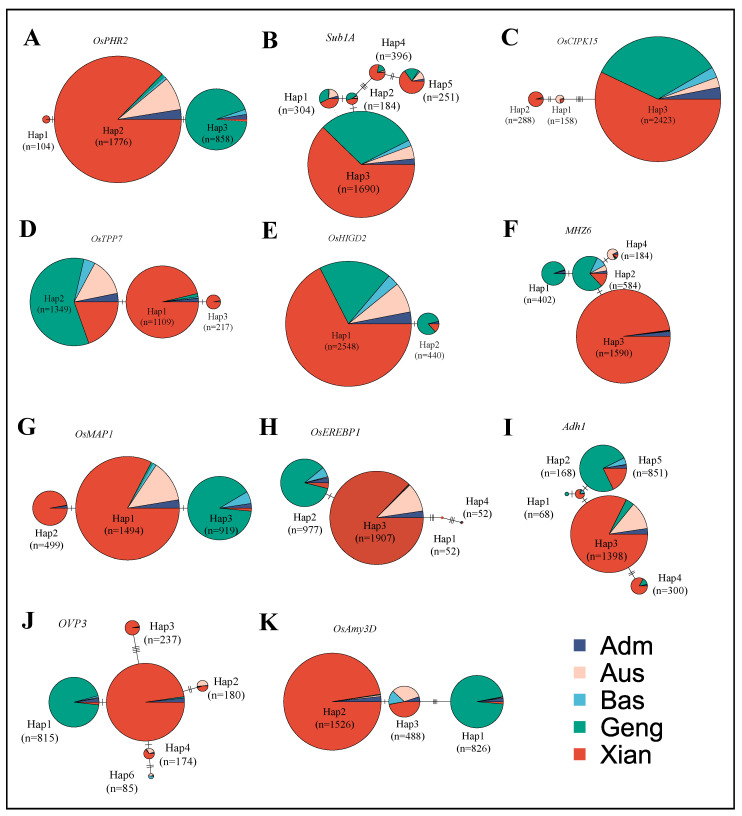
Haplotype network analysis of 11 hypoxia tolerance genes in the 3kRG. Note: n is the sample size. Within each haplotype network, two adjacent gcHaps are separated by mutational changes with hatches indicating differences between the two most related haplotypes. (**A**–**K** are hypoxia tolerance genes *OsPHR2*, *Sub1A*, *OsCIPK15*, *OsTPP7*, *OsHIGD2*, *MHZ6*, *OsMAP1*, *OsEREBP1*, *Adh1*, *OVP3* and *OsAmy3D*, gcHap network analysis in Adm, Aus, Bas, Geng and Xian population).

## Data Availability

Not applicable.

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
