# Peer review of "Identification and Regulation of Hypoxia-Tolerant and Germination-Related Genes in Rice"

_ijms, 2024, doi:10.3390/ijms25042177_

Round 1
Reviewer 1 Report
Comments and Suggestions for Authors
Though this manuscript is categorised as a 'Review', as it includes in silico analyses of genes in response to hypoxia in rice, it is more suitable as a 'Research article', with detailed section on Materials and Methods, Results and Discussion.
LN 47 and elsewhere: use 'genotypes' instead of 'germplasm materials'
LN 241-242, and : could the authors elaborate on the methodology used for screening and functionally categorizing the genes mentioned in Table 2.
LN 263-264: please elaborate on the methodology used to investigate the role of hypoxia tolerance genes in plant metabolic pathways
LN 317: oleoresin steroids?
Figure 2: is not mentioned in the text. Also, please elaborate on the methodology used for data shown in Figure 2.
Reference to be cited: LN 98-100, LN 113-115, LN 139-141, LN 160-163
LN 143: please elaborate on relative damage percent of dry weight (RDP)
Check sentences for grammatical and/or typographical errors, or re-write phrases for clarity: LN 77, 82, 85, 89, 91, 97-98, 123, 133, 129-131, 133-136, 173, 206, 212, 216, 227, 241-243, 268, 283, 285-287, 294, 321-323
Comments on the Quality of English LanguageMinor editing in English language is required.
Author Response
1.LN 47 and elsewhere: use 'genotypes' instead of 'germplasm materials'
Response: We would like to thank the reviewers for their discovery of errors in the text editing errors, thank you very much for your question, We have made changes in the article with the unified name ’genotypes’. With LN47,105;132,152,153,187,614. Thank you again for commenting on this article in your busy schedule.
2.LN 241-242, and could the authors elaborate on the methodology used for screening and functionally categorizing the genes mentioned in Table 2.
Response: We would like to thank the reviewers for their suggestions, which we have revised in detail of in Table 2 with appropriate additions in the article, With LN268-273.
3.LN 263-264: please elaborate on the methodology used to investigate the role of hypoxia tolerance genes in plant metabolic pathways
Response: We would like to thank the reviewers for their suggestions, which we have revised and explained in detail the role of hypoxia tolerance genes in plant metabolic pathways in the article. With LN328-335, LN337,339-345.
4.LN 317: oleoresin steroids?
Response: We would like to thank the reviewers for their discovery of errors in the text editing errors, We modify it to brassinosteroid (BR) which is very important to improve our writing in the future.
5.Figure 2: is not mentioned in the text. Also, please elaborate on the methodology used for data shown in Figure 2.
Response: We would like to thank the reviewers for their discovery of errors in the text editing errors, We have revised and explained this in detail of in Figure 2.in the article and annotated it in the Figure2 note, With LN411-415.
6.Reference to be cited: LN 98-100, LN 113-115, LN 139-141, LN 160-163.
Response: We would like to thank the reviewers for correcting the references in the article, which we have revised in detail in the article reference, LN 98-100, LN 113-115, LN 139-141, LN 160-163, etc.
7.LN 143: please elaborate on relative damage percent of dry weight (RDP)
Response: We would like to thank the reviewers for their suggestions, and which we have explained in detail of RDP in the article. With LN162-163.
8.Check sentences for grammatical and/or typographical errors, or re-write phrases for clarity: LN 77, 82, 85, 89, 91, 97-98, 123, 133, 129-131, 133-136, 173, 206, 212, 216, 227, 241-243, 268, 283, 285-287, 294, 321-323
Response: We would like to thank the reviewers for their suggestions, We have checked the grammar and typography of the sentences and made appropriate additions for clarity: LN: 77, 82, 85, 89, 91, 97-98, 123, 133, 129-131, 133-136, 173, 206, 212, 216, 227, 241-243, 268, 283, 285-287, 294, 321-323, etc.
|
Response to Comments on the Quality of English Language |
|
Point: Minor editing in English language is required. |
|
Response: We would like to thank the reviewers for their suggestions, and we are very grateful for the issues you raised, We carefully check the English language to make English sentences clearer and more fluent and understandable, With LN117,195,196242,252-255,365,368,375-377, etc. Thank you again for commenting on this article in your busy schedules. |

Reviewer 2 Report
Comments and Suggestions for Authors
I have found the article interesting and informative. The following changes are required to be incorporated.
1. The introduction is one long flow. Break it up into paragraphs notably starting from line 52, 62 etc. This will coherently introduce the subject.
2. Also explain why hypoxia understanding is important for rice. Many readers would like to know. It is not mentioned anywhere.
3. In Table2, V-protein, any particular category or miscllaneous. Do include the type of proteins in the main table instead of as footnote.
4. Mostly you have made an inventory of the genes/QTLs etc from the literature. There should be some analysis of the same.
5. Discussion can be made more objective and improved.
6. In perspective bring out the gaps from your review what can be or should be adressed to improve this trait in terms of genes/pathway etc.
Comments on the Quality of English LanguageLot of scope for language improvement to make it coherent and readable.
Author Response
1.The introduction is one long flow. Break it up into paragraphs notably starting from line 52, 62 etc. This will coherently introduce the subject.
Response: We would like to thank the reviewers for their suggestions, and we are very grateful for the issues you raised, We have segmented the introductory section according to the article and added content as appropriate (With LN53,64,90). Thank you again for commenting on this article in your busy schedule.
2.Also explain why hypoxia understanding is important for rice. Many readers would like to know. It is not mentioned anywhere.
Response: We would like to thank the reviewers for their suggestions, We have elaborated and enriched the article with various aspects of the importance of hypoxia for direct seeded rice, With LN39,48-52,95-99.535,549,569, etc.
3.In Table2, V-protein, any particular category or miscllaneous. Do include the type of proteins in the main table instead of as footnote.
Response: We would like to thank the reviewers for their discovery of errors in the text editing errors, We have made detailed modifications and categorized the genes directly in the article and do not explain them as In Table2 notes With LN268-273.
4.Mostly you have made an inventory of the genes/QTLs etc from the literature. There should be some analysis of the same.
Response: We would like to thank the reviewers for their suggestions, We have illustrated the listed genes/QTLs in the article, in terms of pathways, gene-to-gene linkages, and role for direct seeded rice. With LN291-311,315-319,320-322.
5.Discussion can be made more objective and improved.
Response: We would like to thank the reviewers for their suggestions, We describe the importance of hypoxia genes for rice breeding in terms of molecular marker technology, cloning, and Bioinformatics analysis of genes, With LN556-569, 581,582-584,590-596.
6.In perspective bring out the gaps from your review what can be or should be addressed to improve this trait in terms of genes/pathway etc.
Response: We appreciate your questions and ideas for improvement this is very important for our article writing, With LN556-569, 581,582-584,590-596,617-625. thank you again for commenting on this article in your busy schedules.
|
Response to Comments on the Quality of English Language |
|
|
Point: Lot of scope for language improvement to make it coherent and readable. |
|
|
Response: We would like to thank the reviewers for their suggestions, and we are very grateful for the issues you raised, We made major changes to the English grammar, word cohesion, transitions, etc. throughout the article. With LN 117,195,196242,252-255,365,368,375-377, etc. Thank you again for commenting on this article in your busy schedules. |
|

Reviewer 3 Report
Comments and Suggestions for Authors
The present review collects evidence about the response of rice to hypoxia and the genes involved in these processes. I have some minor comments for the authors to improve it:
-Figure 2 is not mentioned in the text. Where were these heatmap taken from?
-the authors should add a section specifically devoted to describe the anatomical modification occurring into the root structure in presence of the stress and also some images of it.
-the part relative to microRNAs is very interesting. First of all I would like that the authors would underline the role of microRNA in plant development and in respose to biotic and abiotic stress; then, they should report (if possible other evidence on it).. please, see and cite the following to do it (Journal of Plant Growth Regulation, 2023, 42.4: 2115-2123;Biochemical and biophysical research communications, 2010, 393.3: 345-349; Plants, 2020, 9.12: 1704).
Author Response
1.Figure 2 is not mentioned in the text. Where were these heatmap taken from?
Response: We would like to thank the reviewers for their discovery of errors in the text editing errors, thank you very much for your question, We have revised and explained this in detail in the article and annotated it in the Figure 2 note, (with LN374-375,376-377,404-406,411-415). Thank you again for taking the time to comment on this article.
2.the authors should add a section specifically devoted to describe the anatomical modification occurring into the root structure in presence of the stress and also some images of it.
Response: Thanks for your suggestions, Your suggestions are very important and useful for improving the quality. Our team is studying on the gene mining and functional analysis of hypoxia tolerance during germination, using natural population and selective introgression lines. Then the root structure in presence of the stress and some images will be conducted with some representative material in the future, but it need a relatively long time. So these contents will be presented in our future papers. Thank you again.
3.the part relative to microRNAs is very interesting. First of all I would like that the authors would underline the role of microRNA in plant development and in respose to biotic and abiotic stress; then, they should report (if possible other evidence on it). please, see and cite the following to do it (Journal of Plant Growth Regulation, 2023, 42.4: 2115-2123; Biochemical and biophysical research communications, 2010, 393.3: 345-349; Plants, 2020, 9.12: 1704).
Response: We would like to thank the reviewers for their suggestions, We emphasize in the text the role of microRNAs in plant growth and development and in response to biotic and abiotic stresses, with appropriate additions for their role in other plants (With LN66-76), Thank you again for commenting on this article in your busy schedules.

Reviewer 4 Report
Comments and Suggestions for Authors
Dear Authors,
As a Review article, it addressed an interesting topic from a scientific and practical point of view. Numerous scientific articles, in concordance to the topic of the study, were consulted.
Methodology of the study was clearly presented, and appropriate to the proposed objectives.
The article has an interesting approach, and the results are presented correctly, in relation to the purpose of the study.
The discussions are appropriate, in the context of the results, and was conducted compared to other studies in the field.
The scientific literature, to which the reporting was made, is recent and representative in the field.
Some suggestions and corrections were made in the article.
The following aspects are brought to the attention of the authors.
1.
Italic Font style for species name
Page 1, row 35
“Oriza sativa L.”
Page 15, row 465
“Arabidopsis”
2.
Please check, is it “R2” or “R2”
Page 5, Table 1
3.
It is appropriate to refer to a figure before presenting it in the content of the article
e.g.
Page 9, Figure 1
Page 13, Figure 4
Similarly to Figure 2, it is recommended to refer to it in the text before it is presented.
Page 11
4.
Please check if it is the ionic form
Page 14, row 420
“H+”
5.
References
It is recommended to provide correct information about each bibliographic source
e.g.
page 17, row 537
“O2” instead of “O<sub>2</sub>”
Row 539
Row 545
Abbreviated Journal Name
"Soil Tillage Res." Instead of “Soil and Tillage Research”
these are just a few examples
It is necessary to review the entire References chapter and make the necessary corrections

Author Response
1.Italic Font style for species name
Page 1, row 35
“Oriza sativa L.”
Page 15, row 465
“Arabidopsis”
Response: We would like to thank the reviewers for their discovery of errors in the text editing errors, thank you very much for your question, We checked carefully and made changes (With LN35,554), which is very important to improve our writing in the future, and thank you again for taking the time to comment on this article.
2.Please check, is it “R2” or “R2”
Page 5, Table 1
Response: We would like to thank the reviewers for their discovery of errors in the text editing errors, We carefully checked that it was “R2”, which is very important to improve our writing in the future, and thank you again for taking the time to comment on this article.
3.It is appropriate to refer to a figure before presenting it in the content of the article
e.g.
Page 9, Figure 1
Page 13, Figure 4
Similarly to Figure 2, it is recommended to refer to it in the text before it is presented.
Page 11
Response: We would like to thank the reviewers for their discovery of errors in the text editing errors, We have carefully revised the content and added Figure 1,2,4. (With LN351,377,450).
4.Please check if it is the ionic form
Page 14, row 420
“H+”
Response: We would like to thank the reviewers for their discovery of errors in the text editing errors, We carefully checked that it was “H+” With LN509.
5.References
It is recommended to provide correct information about each bibliographic source
e.g.
page 17, row 537
“O2” instead of “O<sub>2</sub>”
Row 539
Row 545
Abbreviated Journal Name
"Soil Tillage Res." Instead of “Soil and Tillage Research”
these are just a few examples
It is necessary to review the entire References chapter and make the necessary corrections
Response: We would like to thank the reviewers for correcting the references in the article, We carefully verified references and checked references throughout the text, and at last thank you again for taking the time to comment on this article.
